# GENERATIVE REINFORCEMENT LEARNING WITH TRANSFORMERS

## ABSTRACT

In reinforcement learning, Transformers have been shown to be powerful models for multi-task policy distillation and, to a lesser extent, policy improvement via return interventions within frameworks such as Decision Transformers. These recent results are somewhat atypical for reinforcement learning, as they do not rely on the learning of a value function, which is usually at the heart of most traditional approaches. In this paper, we explore a principled approach to purely generative value function approximation with Transformers, opening the way for existing techniques to be applied for policy improvement. Importantly, unlike other RL methods, this generative approach allows us to kickstart the learning process by fine-tuning strong pretrained state predictors, such as foundation models, substantially shortening the training time. We showcase the potential of our approach by constructing an action-value function for chess that can play at the level of an expert human and over 400 Elo stronger than direct behavioural cloning.

## 1 INTRODUCTION

The Transformer architecture (Vaswani et al., 2017) has caused a massive paradigm shift within machine learning, greatly expanding the scope and capabilities of present-day artificial systems. In the context of reinforcement learning (RL), these models have already had substantial impact, as evidenced by, e.g., Decision Transformers (Chen et al., 2021) or the general-purpose Gato agent (Reed et al., 2022). However, how to best leverage this relatively new architecture in RL is still an open question. At the same time, large-scale pretrained foundation models (Bommasani et al., 2021), based on the Transformer architecture, have become readily available (Touvron et al., 2023a;b), providing a compelling motivation for finding ways to distill their impressive capabilities into RL.

Taking a step back, these recent applications of Transformers are somewhat removed from more traditional value-based RL techniques in that they forgo the need for accurate value-function estimation in favor of directly producing a policy using the sequence modeling capabilities of Transformer-based architectures. These works seemingly call into question the validity of the value function hypothesis (Sutton, 1998), which states that: *All efficient methods for solving sequential decision problems determine (learn or compute) value functions as an intermediate step.* Our goal is to shed light on this discussion point by studying the potential of Transformers to estimate value functions via a generative approach (Veness et al., 2015) to distributional RL (Bellemare et al., 2023) that maps naturally onto the sequence modeling capabilities of Transformers. The key advantage of generative methods for RL is the generality and flexibility that comes with using auto-regressive sequence modeling techniques to deal with both multi-modal, high dimensional, and potentially messy state spaces with atypical structures that make feature-based learning techniques difficult to apply.

**This Work.** We show how to effectively apply the Bayesian decomposition from Veness et al. (2015) with the Transformer architecture to enable robust learning across a variety of domains for both policy evaluation and policy improvement. Importantly, and unlike the other approaches we consider, the generative decomposition allows us to leverage pretrained (foundation) models, typically based on Transformers, to significantly accelerate the process of value function approximation. Fig. 1 provides an overview of the methods we compare in this paper: behavioral cloning (Michie et al., 1990; Schaal, 1996), discriminative distributional RL (Bellemare et al., 2023), and generative distributional RL (Veness et al., 2015).

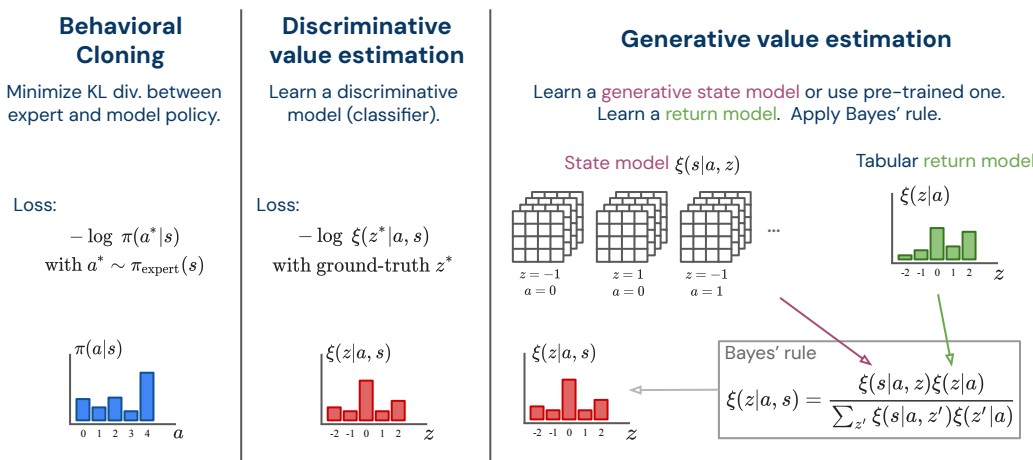

Figure 1: The three methods of estimating the action-value function considered in this paper: *Left.* Behavioral cloning. *Middle.* Discriminative distributional RL (Bellemare et al., 2023), which models the distribution of the return function $\xi(z \mid a, s)$. *Right.* Generative distributional RL (Veness et al., 2015), which models the return function, but by applying Bayes rule and modeling the two (generative) distributions $\xi(s \mid a, z)$ and $\xi(z \mid a)$. The state model $\xi(s \mid a, z)$ can either be trained from scratch, via log-loss minimization, or fine-tuned from $\xi(s)$ using our framework.

**Contributions.** We make the following contributions:

- We develop a novel method for applying Transformers to policy evaluation and improvement through a generative decomposition of the discriminative distributional RL objective.
- We demonstrate that the generative decomposition allows us to leverage pretrained state predictors, thus significantly accelerating the learning process.
- We conduct an extensive empirical evaluation of generative distributional RL with Transformers in the domains of chess (for policy evaluation) and gridworlds (for policy improvement).

## 2 RELATED WORK

Transformers have been successfully applied for large scale policy distillation in a multi-task meta-learning setting across a large number of domains (Reed et al., 2022). The most impressive aspect of the resultant agent is its ability to deal with multiple and diverse input modalities, including free-form text, video game images, and robotics data. Although on a superficial level it may seem that using a probabilistic sequence model in a multi-task setting across a collection of task specific action-observation-reward triples poses no immediate problems, there are subtle issues that need to be considered, which are explored by Ortega et al. (2021). In particular, one needs to take care with respect to masking the input history or ensuring that each task is sufficiently identifiable from the observations to avoid delusions; how to best address this is still under active study. Transformers are also known to be notoriously difficult to apply to RL domains (Parisotto et al., 2020), with architectural changes such as gating showing promise to address this within PPO based approaches. Our work shares some similarities with Decision Transformers, which have received considerable attention recently (Chen et al., 2021; Zheng et al., 2022). The difference with our approach is that we do not rely on return-to-go interventions, but rather frame everything probabilistically and appropriately marginalize over the return space. While Decision Transformers have shown great promise empirically across a wide variety of tasks, it is straightforward to demonstrate theoretically that they are ill-founded in large classes of stochastic environments: e.g. intervening with an impossible desired return will cause the model to condition on an event with measure zero, which can lead to arbitrarily bad behavior.

Recently, the game of chess has gained interest as a testbed to study the capabilities of Transformers. Approaches range from feeding board states directly to Vision Transformers (Czech et al., 2023)

to encoding chess games using natural language (Toshniwal et al., 2022; DeLeo & Guven, 2022) and fine-tuning pretrained models (Noever et al., 2020; Feng et al., 2023). However, none of the prior works using Transformers to build a powerful chess player considered a reinforcement learning perspective, instead working purely in the supervised setting, rendering their results incomparable to the ones in our paper. For our policy evaluation experiments, we use Stockfish (Romstad et al., 2008), a state-of-the-art chess engine, as the stationary policy. Finally, AlphaZero (Silver et al., 2017) is a well-known deep RL algorithm that has achieved remarkable results on chess by combining neural networks with Monte-Carlo tree search (Kocsis & Szepesvári, 2006). However, unlike AlphaZero, in this work, we do not consider search and focus exclusively on value function approximation.

We recall for this paragraph that lossless compression is equivalent to sequence prediction (Delétang et al., 2023). The Compress and Control paper (Veness et al., 2015), that this work is based on, used compressors to directly compress the states and retrieve a probability distribution from the coding lengths. This work demonstrated the applicability of compression techniques to the policy evaluation problem, and to reinforcement learning as a whole. Using compression to aid with machine learning goes back to at least Frank et al. (2000), where compression-based models were compared to classical machine learning methods on natural language problems. The idea of using a generative decomposition for classification was already used in Larochelle & Bengio (2008). More recently, Hamilton et al. (2013) used compression with predictive state representation on domains with large observation spaces to aid with its modelling intractability. Botvinick et al. (2015) discussed the internal representation of RL, specifically a natural and efficient coding of the internal representation.

## 3 BACKGROUND

**Autoregressive Transformer-Based Modeling.** For the purposes of exposition, we need some notation to refer to Transformer-based models that abstracts away many of the underlying implementation details. A Transformer is a probabilistic, context-based prediction model $\rho_\theta : \mathcal{C} \times \mathcal{X} \to (0, 1)$ parameterised by $\theta$, where $\mathcal{C}$ is the context space and $\mathcal{X}$ is the response space. Both $\mathcal{C}$ and $\mathcal{X}$ will be sets of strings of tokens from a common finite set of tokens $\mathcal{T}$, with $\epsilon$ denoting the empty string. As we will deal with a wide variety of different state-spaces, in this work $\rho_\theta$ will always be defined implicitly by using a standard autoregressive decomposition, namely we start with a token-level contextual predictor $\nu_\theta : \mathcal{C} \times \mathcal{T} \to (0, 1)$ and use the chain-rule to define

$$\rho_\theta(x \,|\, c) := \prod_{t=1}^{d} \nu_\theta(x_t \,|\, c \circ x_{<t}).$$

where $x = x_{1:d} \in \mathcal{T}^d$. Here $\circ$ denotes the string concatenation operator and $x_{<t} = x_{1:t-1}$.

**Markov Decision Processes.** A Markov Decision Process (MDP) is a type of probabilistic model widely used within reinforcement learning (Sutton & Barto, 2018; Szepesvári, 2010) and control (Bertsekas & Tsitsiklis, 1996). In this work, we limit our attention to time homogenous MDPs whose action and state spaces are finite. We also focus on finite horizon returns. Formally, an MDP is a tuple $(\mathcal{S}, \mathcal{R}.\mathcal{A}, \mathcal{P})$, where $\mathcal{S}$ is a finite, non-empty set of states, $\mathcal{R}$ is the non-empty set of rewards, $\mathcal{A}$ is a finite, non-empty set of actions and $\mathcal{P}$ is the transition probability kernel that assigns to each state-action pair $(s, a) \in \mathcal{S} \times \mathcal{A}$ a probability measure $\mathcal{P}(\cdot \,|\, s, a)$ over $\mathcal{S} \times \mathcal{R}$. $\mathcal{S}$ and $\mathcal{A}$ are known as the *state space* and *action space* respectively. The transition probability kernel gives rise to the *state transition kernel* $\mathcal{P}(s'|s, a) := \mathcal{P}(\{s'\} \times \mathcal{R} \,|\, s, a)$, which gives the probability of transitioning from state $s$ to state $s'$ if action $a$ is taken in $s$.

An agent's behavior is determined by a *policy*, that defines, for each state $s \in \mathcal{S}$ and time $t \in \mathbb{N}$, a probability measure over $\mathcal{A}$ denoted by $\pi_t(\cdot \,|\, s)$. A *stationary policy* is a policy which is independent of time, which we will denote by $\pi(\cdot \,|\, s)$ where appropriate. At each time $t$, the agent communicates an action $A_t \sim \pi_t(\cdot \,|\, S_{t-1})$ to the environment in state $S_{t-1} \in \mathcal{S}$. The environment then responds with a state-reward pair $(S_t, R_t) \sim \mathcal{P}(\cdot \,|\, S_{t-1}, A_t)$. Here, we will assume that each reward is bounded between $[r_{\min}, r_{\max}] \subset \mathcal{R}$ and that the system starts in a state $s_0$ and executes for an infinite number of steps. Thus, the execution of the system can be described by a sequence of random variables $A_1, S_1, R_1, A_2, S_2, R_2, ...$

The finite $m$-horizon *return* from time $t$ is defined as $Z_t := \sum_{i=t}^{t+m-1} R_i$. The expected $m$-horizon return from time $t$, also known as the *value function*, is denoted by $V^\pi(s_t) := \mathbb{E}[Z_{t+1} \,|\, S_t = s_t]$,

where $\mathbb{E}$ is the expectation with respect to $\pi$ *and* $\mathcal{P}$. The return space $\mathcal{Z}$ is the set of all possible returns. The *action-value function* is defined by $Q^\pi(s_t, a_{t+1}) := \mathbb{E}[Z_{t+1} \mid S_t = s_t, A_{t+1} = a_{t+1}]$. An *optimal policy*, denoted by $\pi^*$, is a policy that maximizes the expected return $\mathbb{E}[Z_{t+1} \mid S_t]$ for all $t$. In our setting, a state-dependent deterministic optimal policy always exists.

**Connecting Value Functions to the Logarithmic Loss.** The main insight with dual reinforcement learning techniques is to learn a time-invariant stationary distribution which contains sufficient information to reconstruct the value function. Recall that the combination of an MDP $\mathcal{M} := (\mathcal{S}, \mathcal{A}, \mu)$ and stationary policy $\pi$ will give rise to a time homogeneous Markov Reward Process over $\mathcal{S}$. Under appropriate conditions, this will give rise to a unique stationary distribution $d_\mathcal{M}^\pi : \mathcal{S} \to [0, 1]$, which is typically referred to as the state-occupancy measure in the literature. Once the chain has been executed for sufficiently long such that all transient effects have disappeared, the quantity $d_\mathcal{M}^\pi(s)$ gives the (limit) probability that the system formed by $\mathcal{M} + \pi$ will be in state $s$ at any given sufficiently large time. Thus a dataset formed by executing the chain and recording the observed states can be considered a sample from $d_\mathcal{M}^\pi$. One might wonder then whether stationary measures over richer objects than merely $\mathcal{S}$ are implied by such a quantity, and whether they can contain sufficient information such that the value function can be recovered from them in a tractable manner? The answer to both those questions is yes, but the details are somewhat tedious; see Section 3.2 of (Veness et al., 2015) for more information.

Given a finite horizon $m$ and the setup above, one can show that a unique stationary measure $\xi$ for the time-homogeneous Markov chain over the *augmented* state-space $\mathcal{S}_{aug} := (\mathcal{A}, \mathcal{S}, \mathcal{R})^{m+1}$ exists. In other words, given a realized interaction sequence of length $n \gg m$,

$$a_1, s_1, r_1, \ldots, a_{m+1}, s_{h+1}, r_{m+1}, \ldots, a_n, s_n, r_n,$$

obtained by executing a stationary policy $\pi$ in $\mathcal{M}$, we can construct a set $\mathcal{I}$ of augmented states in the form

$$\mathcal{I} = \{(a_1, s_1, r_1, \ldots, a_{m+1}, s_{m+1}, r_{m+1}), (a_2, s_2, r_2, \ldots, a_{m+2}, s_{m+2}, r_{m+2}), \ldots,$$
$$(a_{n-m}, s_{n-m}, r_{n-m}, \ldots, a_n, s_n, r_n)\},$$

which for $n \to \infty$ will converge to the stationary distribution $\xi : \mathcal{S}_{aug} \to [0, 1]$. Since the return can be defined as a function of each augmented state, for example, in the finite horizon undiscounted case $z_i := \sum_{j=0}^m r_{i+j}$, we can consider it to also be part of the augmented state. Since $\xi$ exists, then its marginal distribution $\xi(a, s, z)$ also exists, where $a$ and $s$ denote the first action and state in the augmented state space respectively, and $z$ denotes the return. This marginal distribution contains all the information we need to define the value function, and can be marginalized, conditioned, decomposed, and factored in various ways.

**Discriminative and Generative Decompositions.** Here we consider two natural decompositions of the action-value function $Q(s, a)$. First, notice that the probability of getting a return of $z$ starting in state $s$ and executing action $a$ is simply $\xi(z \mid a, s)$. Therefore,

$$Q(s, a) = \mathbb{E}_{\xi(\cdot \mid a, s)}[Z] = \sum_{z \in \mathcal{Z}} z \cdot \xi(z \mid a, s) \tag{1}$$

follows immediately by definition. We refer to Eq. (1) as the *discriminative decomposition* of the action-value function, due to the direct connection to the probabilistic assumptions underlying recent distributional RL formulations. In this decomposition, we learn the distribution $\xi(z \mid s, a)$ directly.

Our second formulation follows by applying Bayes rule to $\xi(z \mid a, s)$. Here one obtains the identity

$$\xi(z \mid a, s) = \frac{\xi(s \mid a, z)\, \xi(z \mid a)}{\sum_{z'} \xi(s \mid a, z')\, \xi(z' \mid a)}. \tag{2}$$

We will refer to this formulation as the *generative decomposition*, as it requires to learn a full state model conditioned on the actions and returns, i.e., $\xi(s \mid a, z)$, but importantly, not a (more complex) state transition model. Note that the return model conditioned on the actions $\xi(z \mid a)$ is usually a much simpler object due to the sizes of the action and return spaces. This generative form was introduced by Veness et al. (2015) by generalizing compression-based classification to the case of policy evaluation. Veness et al. (2015) suggested that when using simple (non-neural) models (such as frequency estimators or Context Tree Weighting by Willems et al. 1995) to approximate $\xi(s \mid a, z)$, the generative decomposition was superior to the discriminative one. Part of our contribution is to revisit this hypothesis in light of the more powerful Transformer architecture.

**Behavioral Cloning.** Given the abundance of data consisting of only state-action pairs but not state-action-return triples, such as YouTube videos, behavioral cloning (Michie et al., 1990; Schaal, 1996) has recently gained interest in the machine learning community (Torabi et al., 2018; Baker et al., 2022). Behavioral cloning sidesteps value function approximation by learning to infer the distribution over future actions (rather than returns) from past behavior. Concretely, it tries to approximate $\xi(a^* \mid s)$, where $a^* = \arg\max Q(a, s)$ is the optimal action for state $s$. In practice, we retrieve this action from a policy $\pi_{expert}$, as shown in Fig. 1.

## 4 METHODS

This section outlines the key technical aspects we found essential for obtaining good performance when using a generative decomposition of the action-value function with Transformers. We will justify each of these findings with corresponding ablation studies in Section 5. Computing the value function in Eq. (1) via the generative decomposition from Eq. (2) relies on two models: the state model $\xi(s \mid a, z)$ and the return model $\xi(z \mid a)$. In practice, we do not have access to either of the two and need to approximate them with their estimates $\hat{\xi}(s \mid a, z)$ and $\hat{\xi}(z \mid a)$, respectively. We now discuss how to chose those estimators.

**Action Conditioning.** For $\hat{\xi}(z \mid a)$, we use a Dirichlet-Multinomial model with concentration parameter $\alpha = 1/2$. As learning an action-conditional return model is far simpler than learning an action-return conditioned state model, this simple approach is sufficient for our purposes. We train this model by counting, such that $\hat{\xi}(z \mid a) := \frac{n_{az} + \alpha}{n_a + |\mathcal{Z}|\alpha}$, with $n_{az}$ being the count of the pair $(a, z)$ and $n_a$ being the count of action $a$.

**Return-Action Conditioning.** Conditioning on the return and action is the only way for the model to actually produce different Q-values for different states and actions. All the information contained in $\xi(s)$, also learned implicitly, is completely irrelevant to Q-value estimation. Therefore, the conditioning method discussed here is of utmost importance for this paper. In the earlier work by Veness et al. (2015), action-return conditioning was implemented by using a separate set of parameters $\theta_{a,z}$ for each possible action-return pair. At the time, the authors justified this as a way of getting around the limited expressive power of the underlying state model, but this is obviously undesirable from a scaling point of view.

Since Transformers have a powerful attention mechanism and ability to deal with relatively large token spaces, it initially seemed natural to implement return action conditioning via simply prepending a return token and an action token to the beginning of the context before autoregressively processing the state. However, we found that this technique failed to work well and that the model was not specializing its predictions sufficiently to different action and return combinations. To address this, we add two new (learnable) embedding matrices, one for the return and one for the action, and add these embeddings with the usual token and positional embeddings of the state. Note that this method of conditioning is good only when we want to retrieve $\hat{\xi}(s \mid a, z)$ for single triplet $(s, a, z)$, which happens during training. However, to obtain the full distribution $\hat{\xi}(z \mid s, a)$ for a given state $s$, e.g., to retrieve the Q-values, we need to run inference over all possible pairs of $(a, z)$, which is inefficient. Training being the harder and longer of the two, we chose this method of conditioning nonetheless.

**Pretraining.** In practice, training $\hat{\xi}(s \mid a, z)$ is related to training $\hat{\xi}(s)$, especially for large state spaces like chess boards, as hinted earlier. This is some base information that must be present in the $\hat{\xi}(s \mid a, z)$ model. Importantly, as a consequence, the Compress and Control framework (Veness et al., 2015) allows us to kickstart the learning process by fine-tuning a strong state predictor, such as a foundation model (Bommasani et al., 2021). The foundation model plays the role of $\hat{\xi}(s)$, and we fine-tune it on conditioned data, i.e., triples $(s, a, z)$, to get $\hat{\xi}(s \mid a, z)$.

**Objective Function.** For the *discriminative decomposition*, the loss is straightforward: $-\log \hat{\xi}(z \mid s, a)$. For the *generative decomposition*, we could minimize

$$-\log \hat{\xi}(s \mid a, z), \tag{3}$$

but this objective does not lead to good results. Concretely, minimizing Eq. (3) recovers a good state predictor $\hat{\xi}(s \mid a, z)$ but a relatively bad return distribution function $\hat{\xi}(z \mid s, a)$ (upon applying Bayes rule), compared to the simpler *discriminative decomposition*. However, since we ultimately care about learning the correct $\xi(z \mid s, a)$, we instead minimize

$$-\log \hat{\xi}(z \mid s, a) = -\log \left( \hat{\xi}(s \mid a, z)\, \hat{\xi}(z \mid a) \right) + \log \left( \sum_{z'} \hat{\xi}(s \mid a, z')\, \hat{\xi}(z' \mid a) \right) \qquad (4)$$

directly. Note that if the state model $\hat{\xi}(s \mid a, z)$ is tabular (i.e., not a Transformer, as in Veness et al. 2015), the two losses are equivalent. However, using Eq. (4) adds degrees of freedom into the modeling as constant factors in the $\hat{\xi}(s \mid a, z)$ or $\hat{\xi}(z \mid a)$ will be canceled out in the fraction. As a result, minimizing this loss will lead to a good return distribution $\hat{\xi}(z \mid s, a)$, but the state predictor $\hat{\xi}(s \mid a, z)$ will no longer approximate the ground truth state distribution, i.e., $\hat{\xi}(s \mid a, z) \neq \xi(s \mid a, z)$. Recovering an accurate state model is not the primary goal of value function approximation and we will not consider this problem in the main experiments of this paper. We discuss potential solutions and do extra experiments in Appendix B.

## 5 RESULTS

In this section, we present our extensive experimental evaluation of the different distributional decompositions of the value function approximation problem with Transformers.

### 5.1 POLICY EVALUATION: CHESS

We first consider policy evaluation, which is the task of approximating the value function induced by a (non-trivial) stationary policy. We consider chess as our environment, as it is challenging due to its large action and state space. We use the Stockfish engine (Romstad et al., 2008) as our stationary policy, associating the returns $z$ with the Stockfish score for a given state-action pair.

**Dataset.** We construct a dataset from $600\,000$ Lichess games from February 2023 (using all possible Elos). For each game, we extract all boards (in FEN notation) as the states $s$ and evaluate all possible legal moves, i.e., the actions $a$, for every board using Stockfish 8 to obtain a score in centipawns. We use a time limit of 50ms per legal move, which corresponds to an Elo of 2667, as shown in Table 1. We convert these centipawn scores into win probabilities, using the formula: $\text{win}\% = 50\% \cdot 2/(1 + \exp(-0.00368208 \cdot \text{centipawns}))$, provided by Lichess (`https://lichess.org/page/accuracy`). Then, to obtain the returns $z$, we discretize the win probability into 16 buckets using quantiles such that $\xi(z)$ is uniform across buckets. We ablate the number of buckets below. Note that $s$, $a$ and $z$ are all integer tokens (with the state being a sequence), which can be fed to a Transformer. As a result, we obtain a dataset $D$ of 1.1 billion $(s, a, z)$ triples in the training set, which we shuffle to avoid overfitting to a particular board state. We train the models on all datapoints in this training set, using a single pass. We evaluate all models on a test set consisting of the first 1 million triples extracted from the Lichess games, which corresponds to roughly $24\,000$ boards.

**Agents.** We assume Stockfish to be the ground truth, and define $\xi^*(z|s, a) := [\![(s, a, z) \in D]\!]$ where $[\![\cdot]\!]$ denotes Iverson indicator brackets, hence $Q^*(s, a) = z$ for the unique $(s, a, z) \in D$ and the discretized win rate $z$ according to the Lichess formula above. $a^* := \arg\max_a Q^*(s, a)$ is the action in board state $s$ with highest Stockfish win rate (or score). We consider two different decompositions of the distributional RL objective (discussed in Section 4): the discriminative given by Eq. (1) denoted $(\hat{Q}_{DV}, \hat{\xi}_{DV})$ and the generative given by Eq. (2) denoted $(\hat{Q}_{GV}, \hat{\xi}_{GV})$. We train them on the losses described in Section 4. Moreover, we also evaluate behavioral cloning on subsampled pairs $(s, a^*)$, where $a^*$ is the $\arg\max$ of all Stockfish scores for the state $s$ as defined above. We train the behavioral cloning agents using the log-loss too, i.e., $-\log \hat{\xi}(a^* \mid s)$, resulting in $(\hat{Q}_{BC}, \hat{\xi}_{BC})$. All the models share the same Transformer architecture backbone (with details provided in Appendix A). They have roughly 36 million parameters, and they are all trained on the full dataset of 1.1 billion boards. Finally, for the generative decomposition, we construct a Bayesian

Table 1: Policy evaluation of different agents in chess. The reported metrics are: ELO (100 ELO difference means twice as likely to win), Kendall-Tau (distance between action ordering), action accuracy (whether the predicted action is in the set of the oracle's best actions), log-losses for the return and action predictions, and the root mean squared error of between the predicted and the oracle's win probabilities. Value approximators do not produce any action distribution, and behavioral cloning does not produce any return distribution, so the respective log-losses are N/A. We report 95% confidence intervals. Elos are computed with 800 games in total, and all other metrics are computed with 28000 test boards. The *discriminative* decomposition yields better results on all metrics (emphasized in bold).

| Agent | ELO | Kendall-Tau | Action accuracy | Action log-loss | Return log-loss | Sqrt L2 loss |
|---|---|---|---|---|---|---|
| Generative Value | $2162 \pm 35$ | $20.4\% \pm 0.3$ | $56.2\% \pm 0.6$ | N/A | $1.25 \pm 0.01$ | $5.90\% \pm 0.01$ |
| **Discriminative Value** | $2254 \pm 34$ | $22.1\% \pm 0.3$ | $58.1\% \pm 0.6$ | N/A | $1.11 \pm 0.01$ | $4.83\% \pm 0.01$ |
| Behavioral Cloning | $1752 \pm 55$ | $7.1\% \pm 0.2$ | $49.1\% \pm 0.6$ | $2.42 \pm 0.03$ | N/A | N/A |
| Oracle Stockfish | $2667 \pm 60$ | $100.0\%$ | $100.0\%$ | $0.$ | $0.$ | $0.$ |
| Stockfish depth 10 | $2431 \pm 43$ | N/A | N/A | N/A | N/A | N/A |
| Stockfish depth 5 | $1885 \pm 47$ | N/A | N/A | N/A | N/A | N/A |

estimate of a Dirichlet-Multinomial model $\hat{\xi}(z \mid a)$ using 100 million triples with concentration parameter $\alpha = 0.5$.

**Main Result.**   Table 1 visualizes the main results. As explained in Section 4, for the *generative* decomposition, we start training $\hat{\xi}(s \mid a, z)$ with a pretrained model $\hat{\xi}(s)$, reusing the parameters of the core of the Transformer (including the embeddings of the state tokens). We give more details and show the difference between using a pretrained model or training from scratch below. We compute all metrics as an average over the whole test set. The action accuracy is $[\![a^*(s) \in \arg\max_{a'} \hat{Q}(s, a')]\!]$, whether the agent's best action is the same as Stockfish, accounting for ties. Kendall-Tau measures the difference in action ordering between the agent and Stockfish. It's a coefficient ranging from -100 (exact inverse order) to 100 (exact same order), with 0 being a random correlation. The L2 loss is $(\hat{Q}(s, a) - a^*(z))^2$, where we recall that $a^*(z)$ is the probability of winning estimated by Stockfish. The return log loss is $-\log \hat{\xi}(z \mid s, a)$. The Elo is measured by playing 50 games for all pairs of agents, including the Stockfish version used for training, plus some Stockfish versions with limited depth (depth 5 and depth 10). We recall that for all Stockfish players, we use 50 milliseconds to evaluate each legal action and then take the best. We add an extra 50 games between our two main contestants, the *generative* and *discriminative* value approximation Transformers. $\binom{6}{2} * 50 + 50 = 800$ games are played in total, and reported in a PGN file. Note that we consider it is a draw if the game lasts 150 plies, and we stop the game there. Similarly, we consider one player has won if its advantage in centipawn is higher than 1500. We then use the BayesELO (Coulom, 2008) on this set of games. The extra Stockfish players allow us to have better accuracy, and a baseline. Indeed, we offset all Elo scores returned by BayesELO such that Stockfish depth 10 has an Elo of 2431, as measured by Ferreira (2013), offset by 200 to (roughly) account for the difference between the version used in our paper (Stockfish 8) and the version Ferreira (2013) used in their paper (Houdini 1.5a 64 bits).

The two main decompositions are close in performance, with the *generative* one being slightly worse than the *discriminative*, in all metrics. The *behavioural cloning* agent performs significantly worse than the two others in terms of Elo, being more than 400 rating points weaker, which roughly corresponds to having a less than 5% chance to win against the generative agents. Also note that the behavioral cloning agent, the Kendall-Tau metric is much worse, likely due to the loss not being incentived to rank the different actions in a state correctly.

**Ablation: Return Discretization.**   One important design choice in the approximation of $\xi(z \mid s, a)$ with a Transformer is the discretization of the returns $z$. As discussed earlier, we use the quantiles of the distribution $\xi(z)$ to bucketize returns such that our estimation $\hat{\xi}(z)$ is uniform across all buckets. However, the number of buckets trades off the accuracy and learning complexity: more buckets means a finer approximation of the win probability, but it also means that the distinction is harder for the model. Moreover, for the *generative* decomposition, we need to sum over all return buckets in Eq. (2), so increasing the number of buckets means increasing the number of estimations per

Figure 2: Main metrics for different numbers of return buckets, computed on 28000 test boards. The more buckets, the more accurate the estimation of the true return value, but the harder it is to distinguish between them. We report 95% confidence intervals.

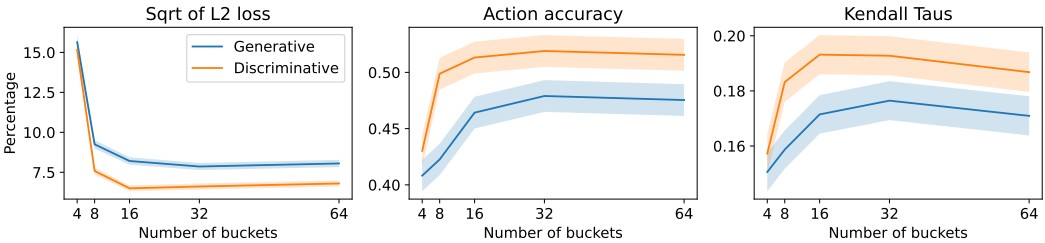

Figure 3: Comparing the learning curves of a model training from scratch or with a pretrained state model $\hat{\xi}(s)$. The losses are computed on a set of 28000 test boards.

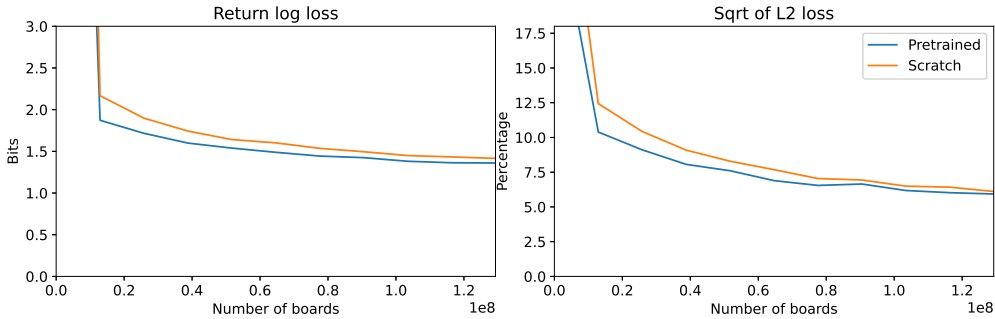

batch and, therefore, the training time. Fig. 2 visualizes the metrics discussed above for a range of different numbers of buckets. Note that we used roughly 10 times smaller models for this sweep, so the absolute numbers are not comparable to the ones in Table 1. We pick 16 buckets in the main paper as a good balance between all the constraints.

**Ablation: Pretraining.** Fig. 3 compares the performance and speed of training a model $\hat{\xi}(s \,|\, a, z)$ from scratch and starting with a pretrained state model $\hat{\xi}(s)$ for the *generative decomposition*. This model was trained on the same dataset of 1.1 billion boards resulting in triples $(s, a, z)$, but without the actions and returns. The loss is therefore simply $-\log \hat{\xi}(s)$. The final loss is $46.6$ bits per board state (described as a sequence of tokens), which are initially encoded as $248$ bits each, leading to a compression ratio of $5.32$. We show some boards with high and low likelihoods using this pretrained model, to have a visual perception of its performance, in Fig. 8.

## 5.2 POLICY IMPROVEMENT: GRIDWORLDS

We now consider the policy improvement problem, which deals with non-stationary distributions. We try our method on simple gridworlds, from Delétang et al. (2021), as a sanity check. We use 3 different levels, taken from Delétang et al. (2021), which are variations of a "catch object" environment, with a reward of one if the agent catches a given object and zero otherwise. Observations are grids of features, integers corresponding to the tile types. This grid, flattened, is the state representation which is then embedded using the trained embedding matrix. Note that the length of the state sequence varies across levels but is contained within the range $[16, 64]$ (the grid sizes varying between 4x4 and 8x8). The returns are the cumulative rewards, and no discount is used, as in the original Compress and Control paper (Veness et al., 2015). We use a finite return horizon $m = 10$, which we enforce by stopping the episode after 10 timesteps.

Figure 4: Average episode returns of policies over 1000 timesteps, trained with the *generative* and the *discriminative* decomposition in three different gridworlds, evaluated using $\epsilon = 0$. Error intervals are the usual 95% confidence interval. We represent the best performing seed out of 3, after training.

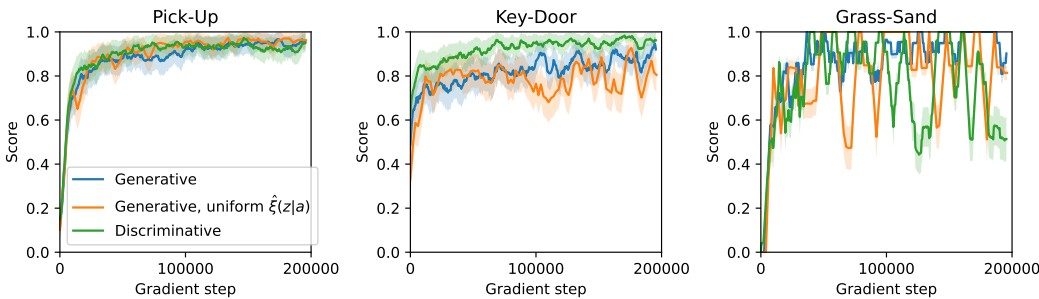

The training is done purely online. The policy is an epsilon-greedy policy over Q-value estimations, with a fixed $\epsilon = 0.2$ throughout training. Each training step consumes a batch of 64 observations and associated observed returns. This batch is provided by the current policy. For both decompositions, we minimize the loss defined in Eq. (4), as in the chess experiments. For the return model $\hat{\xi}(z \,|\, a)$, we use a Dirichlet Multinomial model as explained earlier, but we divide the counts by 2 every thousand batches, to better deal with non-stationarity (Mattern, 2015). We train the models on 200 thousand batches, which corresponds to 12.8 million frames. We use a learning rate of 0.0001. The exact Transformer architecture is described in Appendix A.

Experimentally, it appears as shown in Fig. 4 that using a constant uniform return model $\hat{\xi}(z \,|\, a)$ does not have a big impact on the learning dynamics, and makes the code naturally simpler. This may be an artifact of the return model being mostly uniform in these environments, as the uniform can easily recover from a wrong action. This is still an interesting path to make this method easier to implement in more complex environments.

Both decompositions achieve a return close to 1 per episode after training, on all 3 games. We report the learning curves in Fig. 4. The returns are very noisy due to training purely online.

## 6  DISCUSSION AND CONCLUSION

Taking a step back, it's worth noting the increasing viability of compression based methods to tackle difficult problems in a domain independent way. A Transformer based model (Bellard, 2021) is at the time of writing the best compression technique as assessed by the Hutter Prize competition. Earlier work (Veness et al., 2015) showed that Lempel-Ziv and sophisticated n-gram models derived from Context Tree Weighting could learn to represent value functions sufficient for playing a simple game like Pong; 8 years on, state of the art compression techniques now can play the strategically difficult game of chess at an expert level.

Our results support the Value Function Hypothesis, in the sense that the generative value function approximation techniques produce far better policies than behavioral cloning, but more work is needed to conclusively answer this conjecture. During our case study on chess, we also observed a trend consistent with supervised learning folklore – the superiority of discriminative methods over generative methods on single task setups, though in our case the gap was not large. An exciting next area for investigation is to evaluate these generative value function approximation techniques in a multi-task meta-learning setup, and investigate whether the discriminative or generative decomposition leads to better transfer learning properties. Our pre-training results suggest that the generative decomposition may be more amenable to transfer, but this needs further investigation.

Nonetheless, generative value learning opens a new avenue for fine-tuning pre-trained (foundation) models on complex decision-making problems. We proved that this method works better than behavioral cloning on the game of chess, and is competitive with a more classical discriminative approach, which cannot benefit from fine-tuning.

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

## A   EXPERIMENTAL SETUP

### A.1   TRANSFORMERS

The core architecture we use is always a regular decoder-only Transformer (Phuong & Hutter, 2022). The outputs are always normalized using a log-softmax layer, such that the models output log-probabilities. For the *generative decomposition*, the return and action conditioning is done by adding the embeddings of the return buckets and actions to the state embeddings. They are learnt completely separately of the state embeddings and of each other, but with the same dimension, equal to the dimension of the state embeddings.

In the chess experiments, for the main evaluation and for pretraining, we used the following hyperparameters. 10 layers, with $d_{model} = 512$ (notation from the original Transformer paper (Vaswani et al., 2017)), a widening factor of 4, and 8 heads. The positional encodings are learnt, with a dimension equal to $d_{model}$. The embeddings for the state tokens are also learnt, and also with a dimension equal to $d_{model}$. The total amount of parameters is roughly 36 millions for both decompositions. For the ablations, we used smaller models, similar to the ones used for Atari, which contain roughly 3.6 million parameters.

In the gridworld experiments, we used the following parameters. 4 layers, with $d_{model} = 64$, a widening factor of 4, and 8 heads. The positional encodings are learnt, with a dimension equal to $d_{model}$. The embeddings for the state, the return and the action tokens are also learnt, and also all with a dimension equal to $d_{model}$.

### A.2   CHESS DATASET

The dataset $D$ consists of triples $(s, a, z)$. The states are tokenized chess boards, from FEN strings. It consists in 65 characters=tokens, the first being the side to play (either white 'w' or black 'b'), and the other 64 are the board itself, flattened. Each piece is given a character, using the same mapping as in the FEN strings, but empty squares are attributed the character '.'. Note that we do not flip the board when the side changes: the bottom of the board is always white, and the top is always black, whoever's turn it is to play. Actions are written in UCI format (first square, second square, promotion), for instance 'e2e4' for the well known white opening move or 'e7e8q' for a white promotion to queen. We then enumerate all possible legal actions (using all possible pieces), which gives 1968 possibilities. We arbitrarily associate a number to each action (position in the list), and that becomes the token associated with it. Finally returns are bucketized following quantiles of the distribution $\hat{\xi}(z)$. See next paragraphs.

A natural first investigation of the dataset is to estimate the distributions of states, actions and returns separately, regardless of conditioning. In the following, we use a subset of 100 million triples.

**Estimating $\xi(z)$.**   This distribution is fairly easy to estimate, using a simple count-based model (histogram). We represent this distribution using 50 buckets uniformly spaced over the range [min, max] of the scores. There is a small skew towards negative scores, because there are more actions able to make the player lose the game rather than win the game. Note that there are two peaks at -10000 and 10000, which are the scores we chose to represent checkmates. We also plot the distribution of winning probabilities associated with these returns, between 0 and 100%, all in Fig. 5.

**Estimating $\xi(a)$.**   Actions are represented in the UCI format. There are 1968 possible actions in chess, and obviously many less when conditioned on a given state. To model $\xi(a)$, we also use a count-based model like above. Fig. 6 shows the probability of each action, ordered by decreasing frequency. We also plot the frequencies in log space, but the distribution doesn't seem to follow a power law.

**Estimating $\xi(s)$.**   Finally, we can estimate the probabilities, or rather the log-probabilities given their extremely low value, of the different boards present in the dataset. For this, as explained in the main paper, we train a Transformer model only on the state part of the triples $(s, a, z)$. We plot a histogram of the log-probabilities of the states in Fig. 7. We also show high and low probability states in Fig. 8. Very high probability states are openings: the first peak is the opening position, appearing very often in the dataset.

Figure 5: Histograms of the centipawn scores present in the dataset, and the associated win probabilities. 50 buckets were used for both. The distribution is skewed towards negative scores as we look at all legal actions per board, and most actions are not positive for the player.

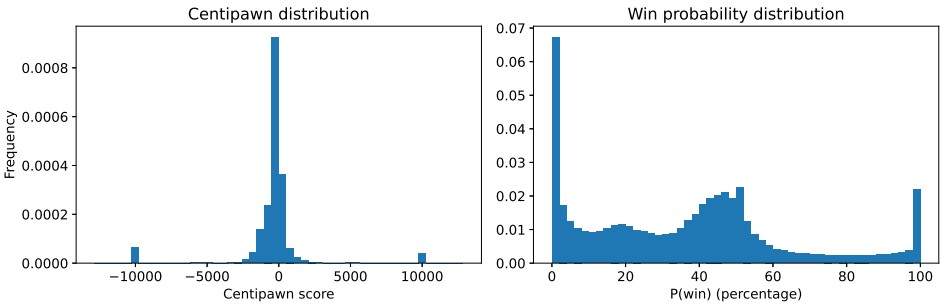

Figure 6: Histogram of the actions present in the dataset, ordered by probability. The 4 most likely actions are [a2a3, h7h6, h2h3, a7a6], which correspond to side pawn opening, a very usual defense against a king side attack.

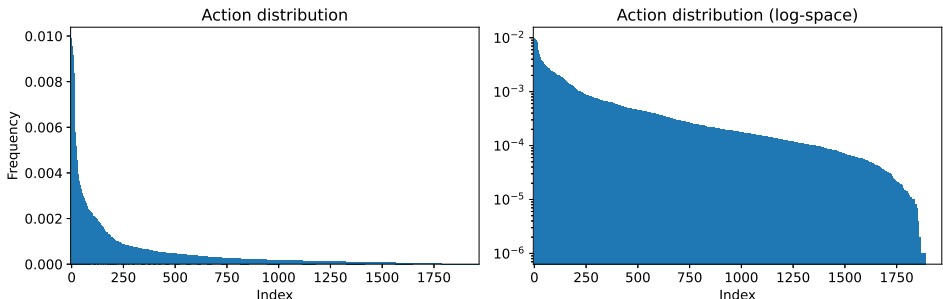

Figure 7: Histogram of the log-probabilities of the states present in the dataset, estimated using a decoder-only auto-regressive Transformer model.

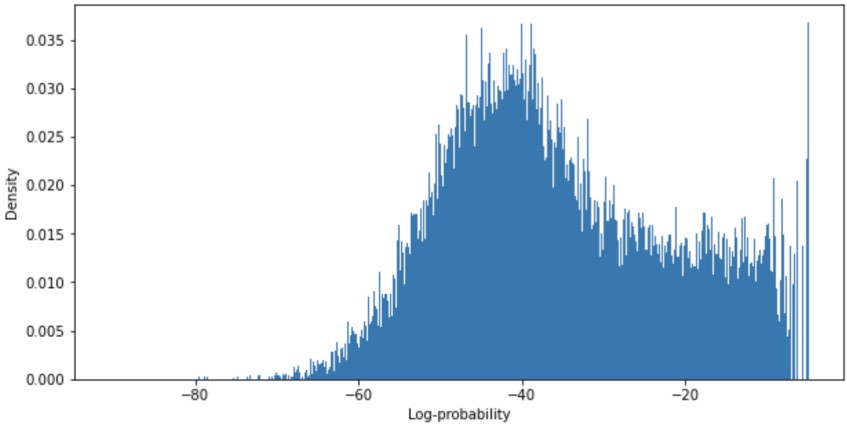

Figure 8: Illustration of the pretrained Transformer model $\hat{\xi}(s)$ on our dataset of chess boards. We pick the least likely and most likely 4 boards of the dataset, using our probabilistic model.

Most likely boards:

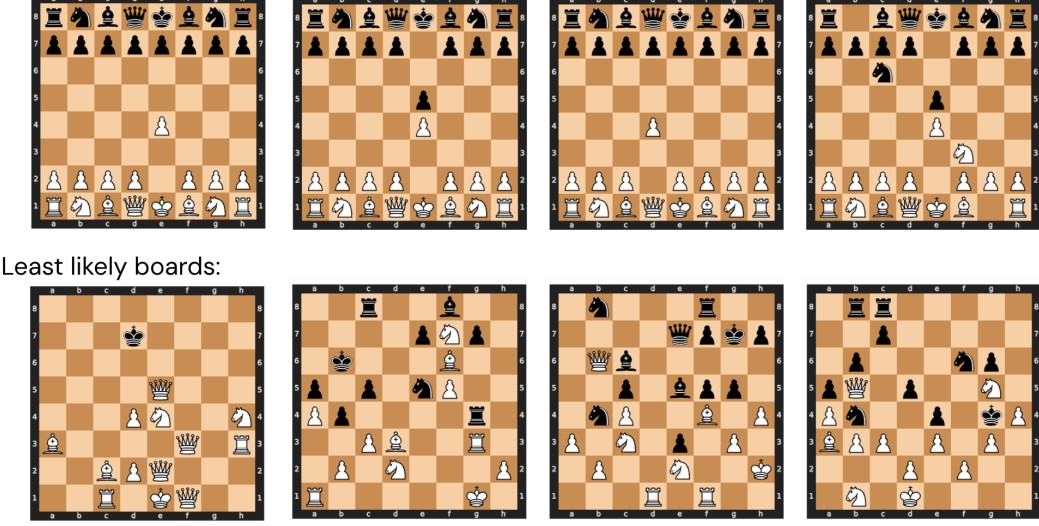

Least likely boards:

## B    ABLATIONS

### B.1    OBJECTIVE FUNCTION

As discussed in the main paper, minimizing the log-loss for the return model (Eq. (4)) naturally increases the loss for the state model (Eq. (3)), which can be undesirable, if we want to recover a good state model $\hat{\xi}(s \mid a, z)$ during, or after training.

To that end, we consider a mixture of Eq. (3) and Eq. (4), parameterized by $\beta \in [0, \infty)$, given by

$$- \log \left( \hat{\xi}(s \mid a, z) \, \hat{\xi}(z \mid a) \right) + \log \left( \sum_{z'} \hat{\xi}(s \mid a, z') \, \hat{\xi}(z' \mid a) \right) - \beta \log \hat{\xi}(s \mid a, z). \qquad (5)$$

This mixture was already introduced in Larochelle & Bengio (2008), in the context of classification tasks. For $\beta = 0$, we recover the return model loss from Eq. (4), and for $\beta = \infty$ we recover the state model loss from Eq. (3). We ablate the hyperparameter $\beta$ in Section 5.

$\beta$ represents a trade-off between a good state model $\hat{\xi}(s \mid a, z)$ and a good return model $\hat{\xi}(z \mid s, a)$. In Fig. 9, we display the various metrics introduced in the previous paragraph for a range of betas, from 0 to 1. In theory, $\beta$ can be much larger, but we found that stopping at 1 was enough as it already leads to catastrophic decrease in return model quality. Overall, as expected, the bigger $\beta$, the higher the L2 and log-losses, the lower Kendall-Tau coefficient, but the lower the state log-loss.

Figure 9: Sweep over various values of $\beta$, in the range [0, 1]. We see that we recover a better state model when $\beta$ increases, at the cost of a worse return model.

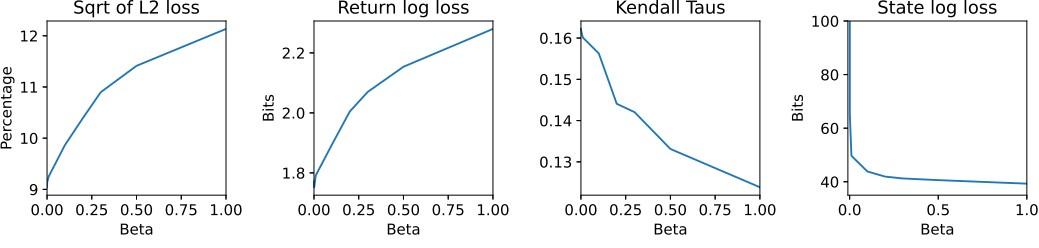

