# OpenReview forum: "Generative Reinforcement Learning with Transformers"
_ICLR.cc/2024/Conference — Submitted to ICLR 2024_

### Official Review · Reviewer_izzx · 2023-10-13

**Soundness:** 3 good
**Presentation:** 3 good
**Contribution:** 1 poor
**Rating:** 3
**Confidence:** 4

**Summary:**

The paper present a method that uses transformers for generative value function approximation in reinforcement learning. The method relies on generative decomposition of the RL objective and on using pre-trained transformers to accelerate the learning process. The authors show experimental results of their method on both policy evaluation and policy improvement.

**Strengths:**

The paper is well written, well presented and the authors have made substantial effort on the experiment side. The contributions appear to be somewhat original although they do built upon the work of Veness et al. 2015. The experimental results are well-presented and conveincing.

**Weaknesses:**

I do not think that the paper has suffecient contributions for ICLR. There are not foundemental contributions on the theory side. The only contribution appears to be on the method side. The authors seem to use foundation models to approximate the state distribution. I am not sure how novel or impactful that is.

On the practical side, it appears that the authors have made quite some effort in the experiments. However, they do not compare their method/results with any similar methods which makes is hard to assess whether there is any improvements compared to the state of the art. Also, for a paper that's such heavilly application-oriented, I'd expect to see clean, well-documented source code in the supplementary material.

**Questions:**

You claim that the objective in eq. 3 does not lead to good results. Is this your finding or known from the literature? Please explain and provide a citation if needed.

At the end of page 4, you say that part of the contribution is to revisit the hypothesis that the generative decomposition is superior to the discriminative one, from Veness et al. 2015. Based on the results shown on Table 1, this hypothesis is not true, right? Please discuss and explain in more detail why that is.

At the end of page 1, you mention that the transformer based architecture accelerates the value function approximation. I did not see any results to backup this claim.

---

> ### Author Response · Authors · 2023-11-22
> **Answers to Questions**
>
> * This was our finding. In our experiments, using this loss with the transformer architecture led to significant training instability, much slower learning, and worse end performance given a fixed training budget. The strongest agent we could obtain using this loss was more than 400 elo worse than using the modified loss. We will include this result for the next revision of the paper, since as you highlighted, it is an important point.
>
> * Our conclusion is that for chess, the discriminative decomposition provides slightly higher performance (around 100 elo in the paper, but that gap has narrowed to 50 since submission with larger models). Our guess is that using a fixed number of parameters, the generative approach suffers from a more difficult learning problem. However, the generative approach benefits significantly from pre-training, so in the end it’s more of a value judgment as to whether you prefer final single domain performance vs data-efficiency. We will add this discussion into the next revision.
>
> * Here we meant to refer to the ability to use pre-training for an improvement in sample efficiency. These results are discussed in Section 5.1, Ablation: Pretraining.

---

### Official Review · Reviewer_A7j8 · 2023-10-27

**Soundness:** 2 fair
**Presentation:** 2 fair
**Contribution:** 2 fair
**Rating:** 3
**Confidence:** 4

**Summary:**

The authors apply the idea of a *generative value function decomposition* (from Veness et al, 2015): Instead of learning the *discriminative* $p(z|s,a)$ for return $z$ directly, they apply the Bayes formula for $p(z|s,a) = p(s|z, a)p(z|a) / \sum\limits_{z'}p(s|z', a)p(z'|a)$ and represent $p(s|z, a)$ by a transformer, and $p(z|a)$ by a simple statistical model.

They claim that this brings the benefit of using pretrained models for $p(s)$ which can be fine-tuned towards $p(s|z,a)$.

Evaluation is done on both policy evaluation and policy improvement on chess.

**Strengths:**

Given the huge success of foundation models, the motivation of being able to use a pre-trained foundation model that can effectively be fine-tuned towards value-estimation is very strong.

**Weaknesses:**

I have several open questions (please see below).

Additionally, the experimental results are interesting, but not convincing w.r.t the strength of the generative model and in particular fine-tuning only seems to provide very marginal gains.

Furthermore, the authors claim that "Our results support the Value Function Hypothesis, in the sense that the generative value function approximation techniques produce far better policies than behavioral cloning".
I don't think BC is the right comparison as it does not make use of the reward at all. In my opinion, a more suitable comparison would e.g. be the decision transformer which still makes use of rewards, but without using a value function as an intermediate step.

**Questions:**

### Clarification

You write that "The key advantage of generative methods for RL is the generality and flexibility that comes with using auto-regressive sequence modeling techniques to deal with both multi-modal, high dimensional, and potentially messy state spaces with atypical structures that make feature-based learning techniques difficult to apply."

I'm not sure I understand what you mean and it seems to be an important part of the motivation for your work; could you please expand on this?

### Architecture

Could you please give a more precise (mathematical?) description of the architecture you are using, in particular how you implement return- and action conditioning as I'm unsure how to understand the explanation given in the text.

### Use of Transformers

Why is the use of transformers in your work necessary? As I understand it, you are learning simply a return-action conditioned density model over states, which could be represented by any network (including e.g. an MLP or a CNN for a suitably encoded state $s$). On the other hand, transformers shine especially for sequence modeling - but as far as I understand the work, you are only modeling individual states, not sequences of states? Please correct me if I misunderstood this.

### On the loss

You claim that if the state model is tabular, the two losses are equivalent. While I'm not sure which two losses you mean, I don't see how it would be equivalent to either of them. Could you please expand on your explanation?

---

> ### Author Response · Authors · 2023-11-22
> **Answers to Questions**
>
> * What we mean here is that any state space can be handled provided it can be broken down into a sequence of tokens. For example in chess, we just provide a fen string representing the board, and the transformer processes it character by character, there is for example no feature construction step required.
> This flexibility shines when dealing with many domains at the same time (e.g. the cited Gato work), which is outside the scope of this work but our ultimate intention.
> * The embedding for a token (in the generative decomposition) is computed as token_embedding[token] + positional_embedding[position of token] + return_embedding[return] + action_embedding[action], as opposed to the usual token_embedding[token] + positional_embedding[position of token] used in NLP. We will make this precise in the next revision in the text using equations, and have committed to releasing the source code if accepted.
> * Transformers were chosen as they are currently the most powerful general compression technique across many benchmarks - our goal was to see how this translated to performance on a challenging domain. One could also use older architectures in principle as you say, with an intermediate feature construction step. The generative approach also modelled the state autoregressively, by using breaking the state (a FEN string) into a sequence of tokens.
> * We are comparing Equation 3 to Equation 4. If for each of the hatted terms, one models them with a frequency based model (i.e. use a naive unsmoothed count based estimate), two interesting things happen. One, Equation 4 = Equation 3, and two, the resultant value estimate is equal to the well-known Every-Visit MC technique for policy evaluation. In this sense you can see CnC as a kind of distributional generalisation of Every-visit MC. We can state this more formally as a proposition in the appendix if accepted.

---

### Official Review · Reviewer_MRfp · 2023-10-30

**Soundness:** 2 fair
**Presentation:** 2 fair
**Contribution:** 3 good
**Rating:** 5
**Confidence:** 4

**Summary:**

This paper tries to compute value functions using transformers (rather than a purely policy-based method as it is the main approach in the recent literature of using transformers for RL). A secondary claim is that the presented method enables using pre-trained models to speed up and improve performance of value learning.

**Strengths:**

Using Bayes rule and learning the components seems to be novel in the context of training transformers for value estimation.

**Weaknesses:**

- The paper spends a lot of space discussing basics (more than 4 pages), then a tiny section on the methods, which is supposed to be the main part. In particular, the section on "Objective Functions" is unclear. The authors have used statements like "X is better than Y" at times with little reasoning. I would strongly suggest to shorten introduction and basics, and add discussions to support such statements, either formal or at least through toy examples with reasonable explanations.

- The chess player agents using supervised learning are still quite important benchmarks to compare against (the authors named them in the second last paragraph of the related work section). Indeed, this is the only way to prove RL is a requirement and rebuttal recent voices that supervised learning reins supreme. Once you learn the value function, its corresponding updated policy (greedy or Boltzmann) can be compared against such agents. Or at the very least, their Elo scores can be compared.

**Questions:**

- Page 4 --> Is it required to have only one communicating class for this discussion to be true? If yes, it needs to be set as an assumption (with a discussion on resultant limitations as most practical domains violate this).

- The counting method used to train the action-conditioned model seems to work only because the MDP is not discounted, otherwise it could be unlikely to have a $n_{za}$ more than 2.

- Page 5, the first paragraph of Return-Action Conditioning --> it is said that all the information in $\xi(s)$ is completely irrelevant to estimating the $Q$ function. It then follows that all such information is also irrelevant to $\xi(s|a,z)$ as well (as it is part of $Q$). Why??

- Then, in the Pretraining paragraph, the authors say: training $\hat{\xi}(s|a,z)$ is related to $\hat{\xi}(s)$ (which is what I expected in the first place). This is contrary to their previous claim.

---

> ### Author Response · Authors · 2023-11-22
> **Answers to Questions**
>
> * Yes, one communicating class is required. The original CnC paper makes an irreducibility assumption on the Markov Chain formed by the combination of MDP+policy, which implies one communicating class. If this holds (along with aperiodicity and positive recurrence), then it also holds for the augmented chain. We can revise the text so as to not sweep this under the rug, but point out that it is a very mild assumption commonly assumed by nearly all real world policy evaluation algorithms which have theoretical guarantees, and far larger than the class of problems for which decision transformers can sensibly address.
>
> * The case of discounted return is more complicated, but the situation is not hopeless. We are aware of two ways of dealing with discounted return. a) Quantize the return, which introduces quantization error; b) Compute the horizon stochastically (e.g. discounting can be seen as the return being geometrically distributed to gamma, so one can keep flipping biased coins to determine an effective horizon for each update, which introduces additional variance to the estimation but provides an unbiased estimate of the discounted return) suitable for a count based estimator. We will provide these details in the next revision.
>
> * Re: Page 5, the first paragraph of Return-Action Conditioning: thankyou for spotting this, this was a very poor exposition on our part, and is completely incorrect. We will remove this clunky sentence in the next revision. What we intended to stress here is that there are multiple ways of implementing conditioning, and we found this to be one of the key challenges in getting the transformer architecture to work well on the challenging domain of chess.

---

### Official Review · Reviewer_vuG2 · 2023-11-01

**Soundness:** 3 good
**Presentation:** 3 good
**Contribution:** 3 good
**Rating:** 6
**Confidence:** 3

**Summary:**

This work shows how to estimate the action value function through both discriminative decomposition and Bayesian generative decomposition with the transformer architecture. The generative approach enables fine-tuning of pre-trained foundational models, resulting in enhancement of learning speed and performance. The study conducted comprehensive empirical experiments in chess for policy evaluation and in gridworlds for policy improvements, showcasing the potential of using the generative approach for value estimation.

**Strengths:**

- Since decision transformer is a goal-conditional offline RL algorithm and the transformer itself acts as a policy. It's natural to question whether value function estimation still has value. This is a pilot work for action value estimation using transformer. It provides insights into the validity of the value function hypothesis given the powerful transformer architecture and the feasibility of finetuning a pretrained foundation model to enhance the learning process and performance.
- Although the generative decomposition idea is not new, previous work only applied to simplified environments. This work demonstrates the efficacy of the discriminative & generative decomposition of the return function (for value estimation) together with the training of transformer in the complex chess environment. It showed an expert-level chess agent can be learned. This work could inspire readers to explore new research in this direction.
- There are several technical details and ablation studies I found to be valuable for practitioners.  Such as the tradeoff of number of return buckets and the way to condition return and action for state. I wish the authors could release the code upon publication.

**Weaknesses:**

1. There is lack of summary of distributional RL in the related work.
2. Page 4, MDP M = (S, A, \mu),  \mu is the transition probability kernel in the CNC paper, should use \mathcal{P} here.
3. The paper focuses on comparing discriminative and generative decomposition of \xi (z | s, a) for estimating the value function. Yet it's not clear how well the value-based approach compared to the decision transformer approach which directly updates the policy.  I suggest adding these experiments for a more comprehensive comparison.
4. The major limitation of this paper is, although it provides some evidence and insights, showing the potential of estimating value function using RL, it's not conclusive whether the value function hypothesis is true. The behavior cloning baseline is too weak to reach a conclusion. Also, the generative approach has not shown advantage over discriminative decomposition, even that it was finetuned on a strong foundation model. This work brings more questions than answers and I do wish there are more answers resolved in this paper.

**Questions:**

1. Figure 3 shows training from scratch does take more training boards to reach the same level of return log loss, but it's not straightforward quantitively how finetuning "substantially shorting the training time". Can you elaborate?
2. In Figure 4, it's said that a constant uniform return model \xi (z | a) does not have a big impact on the learning dynamics. But it seems it causes more oscillation in the Key-Door and Grass-Sand environments and it does have a big impact on dynamics. Could you provide more explanations?
3. Could you run experiments on decision transformers and show how the value-based approach compares to the policy-based approach? If we want to validate the value function hypothesis, shouldn't we collect evidence that the value-based discriminative / generative decomposition is better than policy-based approach?
4. I might miss it, but how does the finetuned generated decomposition approach compared to discriminative decomposition in the experiments?

---

> ### Author Response · Authors · 2023-11-22
> **Answers to Questions**
>
> * We will add some extra references to distributional RL, we omitted many since most works focus on various types of bootstrapping which goes beyond the CnC formulation (which predated distributional RL), but the point is well taken and should be discussed.
>
> * In general one does need to accurately model the xi(z|a) term, however in chess many moves (with the rare exception of promotions) are not good in themselves across arbitrary board states. A helpful example of where this is essential can be found in the Atari game Freeway, where the goal is to move the avatar to the top of the screen, here pressing up/down has a significant effort on the return distribution across nearly all states.
>
> * We take the position that the decision transformer method is a non-starter for the class of ergodic MDPs (especially stochastic environments) and that further demonstration of this is not necessary.
>
> * There is no advantage in terms of final playing strength using pre-training to the generative approach, only shorter training time/sample efficiency. Our latest results suggest the generative decomposition is 50 elo weaker (it was 100 elo weaker in our submission).

---

### Official Review · Reviewer_FxJd · 2023-11-02

**Soundness:** 2 fair
**Presentation:** 2 fair
**Contribution:** 2 fair
**Rating:** 3
**Confidence:** 3

**Summary:**

The paper aims to study the potential of transformers to estimate value function. This is done via a generative approach to distributional reinforcement learning (RL). The approach is evaluated on chess and gridworld environments.

**Strengths:**

* The paper aims to develop a value-based RL method by blending a generative decomposition of distributional RL, which is interesting.
* The paper claims that the proposed method allows the use of pre-trained (foundation) models for value-based RL.
* The method is tested on chess and gridworld environments.

**Weaknesses:**

* The paper feels like a proof-of-concept:
	* The proposed method does not conclusively answer the question asked in the paper (about the Value Function Hypothesis).
	* The proposed method does not beat a baseline (a superiority of discriminative methods was observed).
* Section 3
	* The paragraph "connecting value function to the logarithmic loss" is unclear.
		* The promised connection is not very transparent.
		* A statement claiming that a set of augmented states converges to a stationary distribution is confusing (it seems like a "type mismatch" problem).
	* Section 3, paragraph "discriminative and generative decompositions"
		* The decomposition approach is confusing
			* The main components are $\xi(s|a,z)$ and $\xi(z|a)$. Both are rather unorthodox when it comes to RL: the former requires the knowledge of future returns ($z$), and the latter gives the distribution over returns only given an action (without knowledge of the state $s$).
			* How are such objects used for inference? A pseudo-code could improve the clarity here.
		* Discussion following eq 3 is unclear (e.g., why is $\hat{\xi}(s|a,z)$ good and $\hat{\xi}(z|s,a)$ bad?)
		* There should be an assumption that $|\mathcal Z|<\infty$.
* Experiments:
	* Section 5.1:
		* The evaluation protocol is confusing:
			* Is the computation of ELO metric and early stopping rule described in the "main result" standard? Is there a citation that one can refer to?
			* In the last sentence of paragraph "dataset", it is written that models are evaluated on ~24K boards. Where is it reflected in the paper (Table 1 says about 800 games, roughly 50 for each pair of agents, or computing metrics on 28K boards)
		* Table 1 implies that the generative approach has ELO roughly smaller than 100 when compared to the discriminative approach.  According to the table's caption, the latter is twice as likely to win. This seems to conflict with the statement that "the two main decompositions are close in performance".
		* The results of the ablation part are not very informative (what is the main conclusion? are the conclusions for smaller models also valid for larger ones?). It seems more like content for the Appendix.
	* Section 5.2 (gridworlds).
		* Experiments do not seem very informative, showing that it is impossible to discriminate between the methods significantly.
		* The (online) training protocol could be better explained (e.g., in the form of a pseudo-code).
		* It is curious that using a constant uniform return model does not impact the learning dynamics. It would be interesting to see the results in similar environments where this is not the case.
	* Other:
		* The paper often first describes what did not work and then what was done. For example, eq (3) or the second paragraph in Section 4 "return-action conditioning". While the information about what did not work is very valuable in machine learning, it could be gathered in a separate place and could possibly be put in the Appendix.

**Questions:**

See above.

---

> ### Author Response · Authors · 2023-11-22
> **Answers to Questions**
>
> * Our claim is that our results only support the value function hypothesis in the sense that both value-based methods beat a behavioral cloning baseline. We see our work as more provocative rather than conclusive on this matter.
>
> * Re: A statement claiming that a set of augmented states converges to a stationary distribution: thanks for spotting, this should be “...whose distribution for n->inf will converge to …”
>
> * For inference, we plug the learnt terms into Equation 1 for the discriminative decomposition, and in the generative decomposition we plug the learnt terms first into Equation 2 and then into Equation 1. We will make this more explicit in the next revision.
>
> * Yes, the return space either needs to be finite or quantised.
>
> * An elo gap of 100 implies that the stronger player has a 64% chance of winning, and this is probably what we should have said in the caption. In chess terms, two players with this rating gap would be considered to be close enough to have an enjoyable game with an uncertain outcome, whereas say a 600 rating gap there would be almost no point playing. We can clarify this further in the next revision.
>
> * Using BayeElo for elo computation is standard in the literature. A stopping rule for learnt agent performance is also typically used since degenerate games can take in excess of 800 moves (which are almost always drawn in uninteresting ways), and using it allows for more games in less time, which leads to better elo estimates in practice. We will add some citations to similar setups in the next revision.
>
> * In general one does need to accurately model the xi(z|a) term, however in chess many moves (with the rare exception of promotions) are not good in themselves across arbitrary board states. A helpful example of where this is essential can be found in the Atari game Freeway, where the goal is to move the avatar to the top of the screen, here pressing up/down has a significant effort on the return distribution across nearly all states.

---

### Author Response · Authors · 2023-11-22
**General comments**

We thank all the reviewers for their time and effort.

The main contribution of this paper is to reevaluate various probabilistic decompositions in light of the recent significant advances in sequence modelling brought about by the Transformer architecture. We consider the chess result surprising and significant in its own right, having this level of play produced by a method with no look-ahead search would have been unthinkable 5 years ago.

Regarding a comparison to decision transformers, they have known issues in stochastic and/or nonrecoverable environments, so at best they are a heuristic technique which should be used with caution. In the context of chess, one cannot intervene on a win in a hopelessly lost position, or even more extreme a position where there is not enough material to mate. In terms of our main message, we could for example construct some grid worlds which highlight the problems of decision transformers, but we believe this is already well known in the community. Our chess result in our eyes shows that strong performance is also possible while adopting a principled approach.

If accepted, we will release source code for the chess playing agent. As the reviewers rightly said, this is needed for an application heavy paper.

---

### Meta-Review · Area_Chair_iJyM · 2023-12-05

**Metareview:**

This paper proposes a new approach to generative value function approximation with Transformers, accelerating training by fine-tuning strong pretrained state predictors.

Reviewers generally find that the technical novelty is limited and the experimental results are not convincing.  I agree with these concerns, and hope that the reviews are helpful for improving the paper.

**Justification For Why Not Higher Score:**

Reviewers generally find that the technical novelty is limited and the experimental results are not convincing.  I agree with these concerns.

**Justification For Why Not Lower Score:**

N/A

---

### Decision · Program_Chairs · 2024-01-16

Reject